# Foliar Spraying of ZnO Nanoparticles Enhanced the Yield, Quality, and Zinc Enrichment of Rice Grains

**DOI:** 10.3390/foods12193677

**Published:** 2023-10-07

**Authors:** Sijia Wang, Ruotong Fang, Xijun Yuan, Jie Chen, Kailiang Mi, Rui Wang, Haipeng Zhang, Hongcheng Zhang

**Affiliations:** Jiangsu Key Laboratory of Crop Cultivation and Physiology/Co-Innovation Center for Modern Production Technology of Grain Crops/Innovation Center of Rice Cultivation Technology in Yangtze Valley, Ministry of Agriculture and Rural Affairs, Research Institute of Rice Industrial Engineering Technology, Yangzhou University, Yangzhou 225009, China; 19352670864@163.com (S.W.); 13733758986@163.com (R.F.); yuanxijun135013@163.com (X.Y.); 15665126390@163.com (J.C.); mi877072911@163.com (K.M.); ruiwang0812@163.com (R.W.); hczhang@yzu.edu.cn (H.Z.)

**Keywords:** foliar spraying, head rice, rice quality, zinc oxide nanoparticles, Zn enrichment

## Abstract

Zinc deficiency in rice can lead to reduced nutritional value and taste. This study investigates the potential of zinc oxide nanoparticles (ZnO NPs) as a foliar fertilizer during the jointing stage to improve rice yield, quality, and grain zinc enrichment. Over a two-year field experiment (2019–2020), six doses of ZnO NPs (ranging from 0 to 12 kg hm^−2^) were applied during the jointing stage (46 days after transplanting). The results revealed that foliar spraying of ZnO NPs increased the number of spikelets per spike and the thousand-grain weight by 7.4% to 9.2% and 4.2% to 7.1%, respectively, resulting in a substantial increase in rice yield. Furthermore, it led to a reduction in chalky white and chalky whiteness by 6.23% to 23.6% and 2.2% to 27.9%. ZnO NPs effectively boosted zinc content in rice grains while decreasing the phytic acid to zinc ratio, indicating improved zinc enrichment. Remarkably, protein and amylose content remained unaffected. These findings underscore the potential of ZnO NPs as a foliar fertilizer to enhance rice production, quality, and zinc enrichment. Further research can explore optimal application strategies and long-term effects for sustainable rice production.

## 1. Introduction

Rice serves as a vital staple food for approximately 70% of the global population [1]. It provides a significant portion of the daily caloric intake for billions of people, especially in Asia, where it acts as the primary energy source for countless households. The versatility of rice in various cuisines, along with its compatibility with a wide range of ingredients, positions it as a cornerstone of global culinary diversity. As societies progress and living standards improve, there is a growing consumer demand for high-quality rice. This shift in rice production prioritizes not only high yields but also high-quality rice [2]. Among the various facets of rice quality, consumers are particularly concerned about its nutritional value, as it profoundly influences their health and overall well-being [3]. Micronutrient content, including zinc, has emerged as a critical indicator of rice’s nutritional quality, alongside traditional factors such as protein and starch content [4]. The study of these micronutrients, including their enhancement, has become a key focus in ensuring the health and nutritional security of rice-consuming populations worldwide.

Zinc is an indispensable mineral for human health, and its deficiency can result in various health issues, including reproductive abnormalities, stunted growth, cognitive impairments, compromised immune function, and even susceptibility to tumors. Approximately 20% of the global population, particularly in developing nations, is at risk of inadequate zinc intake. This risk is primarily attributed to the fact that plant-based foods, which constitute a major energy source in developing countries, contain significantly less zinc compared to animal-based foods. Rice, a staple food for millions worldwide, contains an average of 16 mg kg^−1^ of zinc, considerably lower than other plant foods like wheat, corn, and soybeans. Prolonged consumption of rice places individuals at a higher risk of zinc deficiency. Consequently, fortifying rice with zinc through agronomic methods emerges as the most cost-effective, efficient, and practical solution to combat insufficient zinc intake. Moreover, zinc plays a vital role as a micronutrient for plants, participating in numerous critical cellular functions such as enzyme activation, metabolic processes, physiological functions, and ion balance [5]. Nevertheless, due to extensive chemical fertilizer use and years of tillage, soil zinc levels have not been adequately replenished, with more than half of the soils in cereal-growing regions globally experiencing zinc deficiency [6]. Soil zinc deficiency can impede zinc absorption by plants and hinder their growth, ultimately leading to reduced yields and compromised quality in severe cases [7]. The direct application of zinc fertilizer offers a means to provide plants with sufficient zinc for uptake and accumulation in edible parts, making it the most widely adopted and effective agronomic approach for enhancing zinc content in rice [8].

Numerous studies have provided compelling evidence that the judicious application of zinc fertilizer can result in increased rice yields, enhanced rice quality, and elevated zinc levels in rice grains [9,10,11,12]. However, soil application of zinc fertilizers is subject to the influence of various factors, including soil pH, texture, and organic matter content, and is susceptible to immobilization through soil adsorption. A significant proportion of the zinc absorbed by plants is retained in non-edible parts such as roots, stems, and leaf sheaths, resulting in limited zinc bioavailability in the final harvested product. In contrast, foliar spraying represents a method direct application to plant leaves, bypassing soil-related issues and facilitating more efficient zinc absorption by rice plants, with subsequent transfer to the grains, thereby achieving effective zinc enrichment [12]. Several innovative foliar zinc fertilizers, including ZnSO_4_, Zn-EDTA, and Zn-Gly, have been developed and implemented. These fertilizers exhibit high absorption efficiency and significant potential for zinc fortification. However, they still exhibit certain limitations, including low utilization rates, inadequate leaf nutrient adherence, considerable environmental impacts, and the potential to induce leaf burning [13].

Zinc nanoparticles have garnered significant attention in agricultural applications owing to their expansive specific surface area and active surface characteristics, which facilitate crop absorption and utilization [14,15]. Research has elucidated that the judicious application of nano-zinc at appropriate concentrations can foster soil microbial diversity and activity, resulting in enhancements to rice grain quality [16]. Furthermore, ZnO nanoparticles have demonstrated the capacity to stimulate plant growth, mitigate cadmium toxicity, bolster disease and salt resistance in crops, and generally contribute to improved crop performance [17,18,19]. However, the majority of extant studies concerning the utilization of Zn nanoparticles in agriculture have primarily centered on augmenting crop resistance and facilitating zinc uptake by plants. There has been relatively less focus on evaluating their effects on rice yield and quality, particularly with regard to zinc enrichment in rice grain such as brown rice and polished rice. This study addresses this gap by applying different dosages of ZnO nanoparticles to high-quality, palatable japonica rice at the jointing stage. The objective is to comprehensively investigate the influence of these nanoparticles on rice yield, quality, and zinc enrichment in rice grains. The findings of this research endeavor are anticipated to provide both a theoretical foundation and practical insights for the agronomic utilization of ZnO nanoparticles in rice cultivation.

## 2. Materials and Methods

### 2.1. Experimental Materials

The rice variety used in this study was Nanjing 9108, a commonly cultivated rice variety in the middle and lower regions of the Yangtze River. This rice variety has a growth cycle spanning 150–155 days. The seeds were sown on 18 May 2019, 19 May 2020 respectively. After 25 days of growth, the seedlings were transplanted into the field at a density of 133.33 × 10^4^ hills per hectare on 12 June 2019, 13 June 2020 respectively. Harvesting took place on 19 October 2019, 20 October 2020, corresponding to the respective planting dates.

In this study, a randomized block experiment was conducted using plot sizes of 10 m × 6 m, resulting in a total area of 60 m^2^ for each plot. A control group was sprayed with water (referred to as CK), while five different dosages of zinc oxide nanoparticles (ZnO NPs) were applied at rates of 0.75 (T1), 1.50 (T2), 3.00 (T3), 6.00 (T4), and 12.00 (T5) kg per hectare (hm^2^). Each treatment was replicated three times, resulting in a total of 18 blocks. The ZnO NPs used in this experiment were procured from Shanghai Chaowei Nanotechnology Co., Ltd., located in Shanghai, China. These nanoparticles were confirmed to be spherical in shape with an average size of approximately 50 nm, as determined through SEM and TEM characterization (refer to Figure 1 for details). To prepare the ZnO NP solutions, 4.5, 9.0, 18.0, 36.0, and 72.0 g of nanometer-sized zinc oxide were weighed and added to ultra-pure water to create 1 L solutions. These solutions were thoroughly dissolved using ultrasound for 30 min before being applied to rice plants during the gestation stage, specifically on clear, windless days.

### 2.2. Experimental Site

The research was conducted at the Experimental Farm of Yangzhou University (YZU), situated at coordinates 32°23.4′ N and 119°25.2′ E in Jiangsu Province, China. This study spanned two rice growing seasons in 2019 and 2020. The soil used in the experiment had a sandy loam texture and contained 24.40 g kg^−1^ of organic matter, 1.30 g kg^−1^ of total nitrogen, 104.20 mg kg^−1^ of available nitrogen, 35.40 mg kg^−1^ of available phosphorus, and 72.50 mg kg^−1^ of available potassium. The soil had a pH level of 6.51. The experimental site experiences a subtropical monsoonal climate characterized by ample rainfall and mild temperatures. The average annual temperature in this region is approximately 15.2 °C, with an annual precipitation of around 1020 mm. Figure 2 provides meteorological data recorded during the rice growing season.

### 2.3. Agronomic Management Systems

A total of 270 kg hm^−2^ of nitrogen was administered in this study. This nitrogen was divided into three applications: 30% was applied during the transplanting period, another 30% was applied as tillering fertilizer seven days after transplantation, and the remaining 40% was applied as panicle fertilizer. Urea, which contains 46% nitrogen, served as the nitrogen source. Additionally, all treatment groups received phosphorus and potassium at rates of 135 and 270 kg hm^−2^, respectively. These nutrients were supplied in the form of calcium superphosphate and potassium chloride during the initial fertilizer application. The overall crop management practices, including irrigation, disease control, pest control, and weed control, adhered to the guidelines provided by local government agencies.

### 2.4. Sampling and Data Collection

#### 2.4.1. Yield and Yield Components

Upon reaching maturity, the rice from each plot was harvested and underwent manual threshing. Subsequently, it was sun-dried until it reached a moisture level of approximately 14% to determine the grain yield. The number of panicles per hill was determined by counting the panicles from 20 randomly selected rice plant hills across each plot. Additionally, representative samples were collected from 12 hills of rice plants for the purpose of measuring and calculating spikelets per panicle, the seed-setting rate, and the 1000-grain weight.

#### 2.4.2. Grain Quality

Grain quality analysis adhered to the national standards of the China (GB/T 17891-2017) [20]. Following harvest, rice grains from each plot were sun-dried and stored at room temperature for at least one month to prepare for grain quality analysis. A 100 g sample of rice grains was subjected to polishing using a dehusker, resulting in the separation of broken and unbroken grains. The brown rice rate, milled rice rate, and head rice rate were expressed as percentages relative to the total (100 g) rice grains. To assess the chalky rice rate and the degree of chalkiness, a rice appearance scanner (ScanMaker i800, Microtek, Shanghai, China) was employed. Protein content was determined utilizing an automatic Kjeldahl apparatus (Kjeltec8200, Foss, Hillerød, Denmark). The amylose content was measured using the method outlined by Zhu et al. (2017) [21] via iodine colorimetry at a wavelength of 620 nm, with reference to a standard mixture of potato starch. The pasting properties of the rice flour were assessed using a rapid viscosity analyzer (RVA, Super3, Newport Scientific, Warriewood, Australia).

#### 2.4.3. Zinc Content of Polished Rice

A 0.1 g dried sample was carefully weighed and deposited into an Abe bottle. Subsequently, it was subjected to ashing in a muffle furnace at 480 °C for approximately 14 h. After cooling, 2 mL of 15% HNO_3_ solution (imported nitric acid) was introduced, and the mixture was allowed to stand for a duration of 24 h. Following this period, 8 mL of ultra-pure water was added, and the mixture underwent filtration using slow-speed filter paper. The resulting filtrate was subjected to dilution, and the zinc content was quantified employing an iCAP 6300 (Thermo Fisher Scientific, Waltham, MA, USA) atomic absorption meter.

#### 2.4.4. The Phytic Acid (PA) Content and Molar Ratio of PA to Zn in Polished Rice

The method employed to determine the phytic acid content was adapted from Vaintraub and Lapteva’s (1988) [22] protocol with some modifications. Initially, 0.25 g of the sample was combined with diluted hydrochloric acid, subjected to shaking, and then centrifuged to acquire the supernatant. Subsequently, the supernatant was treated with chromogenic agents (FeCl_3_ and sulfosalicylic acid) and compared against a standard solution prepared with sodium phytate. The absorbance value was measured at 500 nm, enabling the calculation of the rice’s phytic acid content. To determine the molar ratio of phytic acid to zinc, the millimoles of phytic acid were divided by the millimoles of zinc.

#### 2.4.5. Statistical Analysis

The data were presented as mean ± SD. Statistical analyses were conducted using SPSS Statistics software (Version 26.0, IBM, Armonk, NY, USA). One-way analysis of variance (ANOVA) was used to determine statistically significant differences (*p* < 0.05) among treatment means.

## 3. Results

### 3.1. Yield and Yield Components

Table 1 displays the impact of ZnO NPs on rice yield and its components. Application of ZnO NPs through foliar spraying resulted in a significant increase in grain yield by 2.3% to 4.1% in comparison to the control (CK). Figure 3 illustrates the growth status of field rice during the grain-filling stage after foliar spraying with ZnO NPs at the jointing stage, confirming the results related to grain yield. During the two-year study, the augmented yield of rice plants treated with ZnO NP spraying could be attributed to increase in the number of spikelets per spike (7.4% to 9.2%), grain filling rate (1.7% to 4.3%), and 1000-grain weight (4.2% to 7.1%). Nevertheless, the ANOVA results revealed that the variation in panicle number was not significantly affected by nanoscale zinc spraying.

### 3.2. Processing Quality

Table 2 illustrates the impact of ZnO NP application on rice grain quality. The application of ZnO NPs at the jointing stage resulted in a significant improvement in the brown rice rate, milled rice rate, and head rice rate. The brown rice rate showed a parabolic trend with an increase in ZnO NPs application amount, and T4 values were the highest at 86.49 percent and 86.19 percent during the two-year experiment. The milled rice rate did not display a significant trend in 2019, whereas it exhibited an increasing trend in the 2020 trial with the rise in the amount of ZnO NPs sprayed. The head rice rate patterns varied slightly in the two-year experiments with increasing ZnO NPs application. In 2019, the head rice rate displayed an increasing trend followed by a decreasing trend, and T4 produced the highest value. However, no deterioration in head rice rate was observed in the 2020 trial.

### 3.3. Appearance Quality

Table 3 indicates that the ZnO NPs treatments had a positive impact on the appearance quality of rice, as evidenced by the length to width ratio of rice, chalkiness grain rate, and chalkiness degree. The length to width ratio of rice showed an increasing trend up to T4, beyond which it decreased. Conversely, the chalkiness grain rate and chalkiness degree exhibited a decreasing trend up to T4, beyond which they increased. The results suggest that the appearance quality of rice may deteriorate when the amount of ZnO NPs exceeds that of T4.

### 3.4. Nutritional Quality

The influence of spraying ZnO NPs on rice protein content and amylose content is presented in Table 4. The protein content of rice demonstrated a parabolic trend with the increase of ZnO NPs, but the change was not statistically significant in comparison to CK. The impact of ZnO NPs treatment on rice amylose was not significant except for T5 in the 2020 trial, which resulted in a significant reduction of 9.8% in rice amylose content compared to CK.

### 3.5. Taste Quality

Table 5 presents the effect of different levels of ZnO NPs on the taste value of rice. The taste value of rice increased initially and then decreased with the increasing amount of ZnO NPs, reaching the highest value in T4, with 82.37% and 81.67% respectively. The T2–T5 treatments showed a significant improvement in rice taste value by 8.9–13.0% compared to CK. The improvement in taste value was mainly attributed to the enhanced appearance value, viscosity value, and balance value of the rice, as well as the reduced hardness value of the rice. However, when the amount of ZnO NPs exceeded the T4 treatment, the hardness of the rice increased, resulting in a decrease in overall taste.

### 3.6. Steaming Quality

The RVA profile is commonly used to evaluate the taste and steaming quality of rice. Table 6 shows that the rice breakdown value increased first and then decreased as the amount of ZnO NPs sprayed increased, with the highest value achieved at the T4 treatment. The RVA measures both the force applied and the distance the probe travels as it penetrates the rice grain. The rice breakdown value signifies the force required to cause the rice grain to break apart or disintegrate. Remarkably, the application of ZnO NPs substantially increased the rice breakdown value by 31.0% to 41.7% compared to the control (CK). This suggests that the rice grains treated with ZnO NPs were more prone to breaking apart under applied force. Sstback value pertains to changes in viscosity observed in a rice paste after it’s been heated and subsequently cooled. It sheds light on the tendency of cooked rice to become firmer upon cooling. In the study, rice sprayed with ZnO NPs exhibited varying degrees of setback value reduction compared to the control (CK). Notably, significant decreases were noted in treatments T2 and T5 during the 2019 trial and in treatments T1–T5 during the 2020 trial. Upon closer analysis, it was found that the increase in rice breakdown value and the decrease in setback value were primarily driven by a reduction in trough viscosity. While peak viscosity and final viscosity showed some fluctuations, they had a comparatively minor impact on the breakdown value compared to the changes in trough viscosity.

### 3.7. Zinc Content of Brown Rice and Polished Rice

The application of foliar sprays of ZnO NPs at the jointing stage significantly increased the Zn content in both brown rice and polished rice compared to the control group (CK). The Zn content in brown rice and milled rice displayed a parabolic trend with increasing levels of ZnO NP application, reaching the highest value in T4 (as shown in Figure 4 and Figure 5). Specifically, the Zn content in brown rice and polished rice increased by 46.4–82.4% and 45.1–79.4%, respectively, indicating that foliar spraying of nano-Zn is an effective method to improve the Zn content of rice edible parts.

### 3.8. The Phytic Acid to Zinc Molar Ratio

The impact of spraying ZnO NPs during the jointing stage on the bioavailability of zinc was also demonstrated through the molar ratio of phytate to zinc in brown rice and polished rice (as shown in Figure 6 and Figure 7). The molar ratio of phytic acid to zinc in brown rice and polished rice significantly reduced by 31.4–43.5% and 30.6–42.9%, respectively, when treated with ZnO NPs. Moreover, the molar ratio of phytic acid to zinc in brown rice and polished rice showed a similar trend, decreasing first and then increasing with the increase in ZnO NP levels, and reaching the lowest point at T4. These results suggest that the use of nano-Zn sprays can effectively enhance the bioavailability of zinc in brown rice and polished rice.

## 4. Discussion

### 4.1. Effects of ZnO Nanoparticles Spraying on Rice Yield

The interaction between leaf physiology and the application of ZnO NPs has profound implications for rice production. Flag leaves, which exhibit heightened physiological activity during the middle and late stages of rice growth, are integral to yield and grain quality [23]. Our study has unveiled that ZnO NPs can exert a positive influence on rice yield through multifaceted mechanisms. Specifically, we observed that ZnO NPs facilitate spikelet formation per panicle, enhance grain filling rates, and increase the 1000-grain weight. These findings align harmoniously with prior research highlighting the phenomenon of ‘N-Zn mutual promotion’ during the late stages of nutrient transport [24]. Additionally, we found that an ample supply of nitrogen can suppress spikelet degradation and further bolster grain filling, underscoring the complex interplay of nutrients in rice development. Moreover, Zn, as a trace element, was found to have a considerable impact on the synthesis of pivotal growth factors such as arginine, glycine, and tryptophan. These growth factors are instrumental in the development of plant embryos and glumes [25]. Our study revealed that moderate increases in Zn fertilization can stimulate floret differentiation and result in a greater number of grains per spike. This observation hints at the potential for optimizing Zn fertilization strategies to maximize rice yield and quality. The intricate relationship between flag leaf physiology and ZnO NPs application underscores the importance of nutrient management in modern rice cultivation.

### 4.2. Effects of ZnO Nanoparticles Spraying on Rice Quality

Rice quality stands as a pivotal determinant influencing its market worth, making it a paramount concern for both producers and consumers. In our investigation, we unearthed compelling evidence regarding the positive effects of foliar ZnO NPs application in comparison to conventional water spraying. This novel approach yielded substantial improvements in multiple vital quality parameters. We observed a noteworthy increase in the brown rice rate, indicating a higher proportion of whole grains with the bran layer intact. Additionally, the milled rice rate, representing the percentage of polished rice obtained from brown rice, exhibited significant improvement. A higher head rice rate was also achieved, indicating an increase in the yield of intact, unbroken grains—a desirable trait in rice quality. Furthermore, the length-to-width ratio of rice grains, an important aspect of appearance and overall rice quality, showed remarkable enhancement. Moreover, our study unveiled a significant reduction in both the chalkiness grain rate and the degree of chalkiness. These are crucial quality indicators, as chalkiness affects the visual appeal and milling quality of rice. These pronounced improvements in rice quality can be attributed to various interrelated factors. Firstly, foliar ZnO NPs application appeared to enhance the efficiency of photosynthetic material transport, facilitating the movement of vital nutrients within the plant. This, in turn, resulted in a more substantial nutrient enrichment within the grains. Additionally, the application led to more uniform distribution of assimilates within the rice plant and denser grain structure. These combined effects not only enhance the overall appearance of the rice but also contribute to improved milling resistance. These findings align with previous research by Yang et al. (2021) and Dimkpa et al. (2020) [26,27], highlighting the multifaceted benefits of ZnO NP application in rice cultivation. Altogether, this interaction effect highlights the potential of foliar ZnO NPs application as a valuable strategy for enhancing rice quality, which could have far-reaching implications for both producers and consumers in the rice market.

Additionally, our study demonstrated that the application of ZnO NPs had a notable impact on the taste quality of rice. It was found to reduce the hardness of rice and enhance its appearance, viscosity, and balance. This enhancement can primarily be attributed to the role of zinc in regulating flavor enzymes in rice. Zinc facilitates the hydrolysis of flavor substances during the steaming process, ultimately resulting in an improved taste profile [27,28,29]. The improvement in taste quality was further reflected in the RVA parameters, a commonly used tool for evaluating rice starch’s gelatinization properties and taste value [30]. Specifically, the breakdown value measured the ease of disintegration of swollen starch granules. Previous studies, such as the one by Yin et al. (2021) [31], have established a positive correlation between breakdown and rice viscosity. The setback value, on the other hand, indicated the tendency of starch pastes to retrograde, serving as an index of starch retrogradation [32]. A lower setback value indicated better cooking quality since the rice did not become excessively hard upon cooling [33]. The observed changes in viscosity and hardness of cooked rice, following ZnO NPs treatment in our experiment, align with these findings. However, it’s worth noting that in contrast to previous research, which found that Zn fertilization significantly increased rice protein content, the rice protein content and straight-chain amylose content in our ZnO NPs treatments fluctuated, and the overall differences did not reach statistical significance. This suggests that ZnO NPs may have a unique impact on these parameters compared to traditional Zn fertilizers.

### 4.3. Effects of ZnO Nanoparticles Spraying on Zn Content in Edible Parts

Efficient protocols for applying zinc may increase the zinc content in grains but don’t necessarily guarantee higher zinc bioavailability in the final food products derived from them [12]. Brown rice and polished rice, as two of the most popular consumer-oriented rice-based food products, were of particular interest in this study. Our findings demonstrated that foliar sprays of ZnO NPs during the jointing stage had a noteworthy impact. Specifically, they led to an increase in the zinc content within both brown rice and polished rice. Additionally, they reduced the phytic acid to zinc molar ratio in the edible parts of the rice grains. This is significant as phytic acid, primarily found in the non-edible parts of the grain, can limit the bioavailability of mineral nutrients in plants, including zinc [34]. A lower phytate to zinc molar ratio indicates higher bioavailability of zinc [35]. Our results indicated that ZnO NPs spraying during the gestation period effectively decreased the antagonism that typically exists between phosphorus and zinc during grain filling. This reduction, in turn, led to a significant decrease in the overall phytic acid to zinc molar ratio in both brown rice and semolina. Although ZnO NPs increase the phytic acid content in these parts, the greater increase in zinc content resulted in a significant overall improvement in the final bioavailability of zinc. This suggests that ZnO NPs can offer a promising approach to enhancing the nutritional quality of rice products.

## 5. Conclusions

The findings in this study highlight the benefits of applying ZnO nanoparticles during the rice spikelet stage. This application improved rice yield and quality while effectively increasing the zinc content in the edible part of the rice kernel. Specifically, the use of ZnO NPs resulted in increased spikelets per spike, higher grain filling rates, and greater 1000-grain weights, all contributing to higher yields. Furthermore, it significantly enhanced rice processing, appearance, flavor, and cooking qualities. Additionally, the application of ZnO NPs led to a notable increase in zinc content in both semolina and brown rice, along with a reduction in the phytic acid to zinc molar ratio. This indicates that zinc enrichment in the edible part of rice grains was effectively achieved. The optimal amount of ZnO NPs spray during the rice jointing period was estimated to fall within the range of 3 to 6 kg hm^−2^.

## Figures and Tables

**Figure 1 foods-12-03677-f001:**
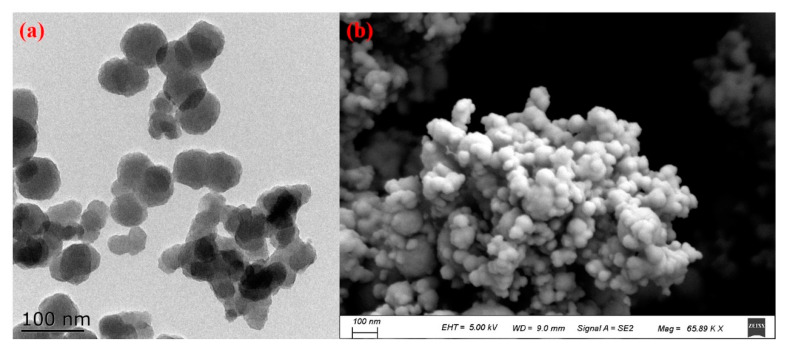
SEM (**a**) and TEM (**b**) images of ZnO NPs.

**Figure 2 foods-12-03677-f002:**
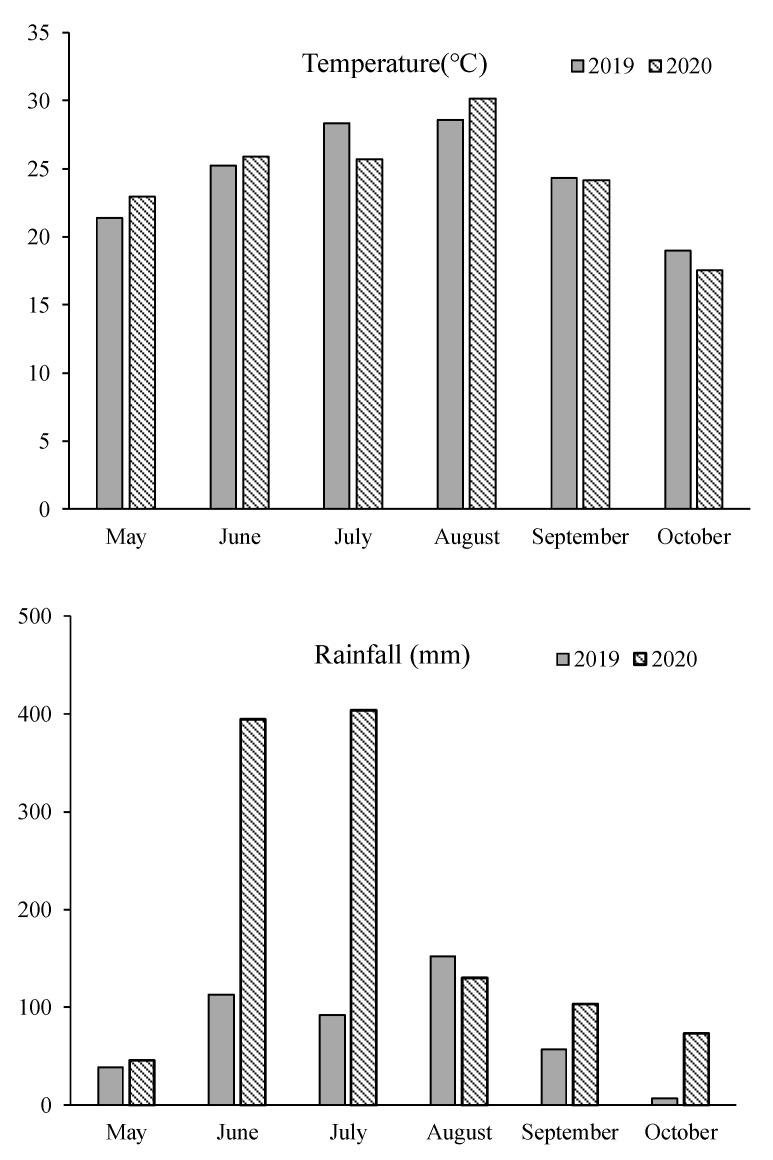
Meteorological data of rice growing season.

**Figure 3 foods-12-03677-f003:**
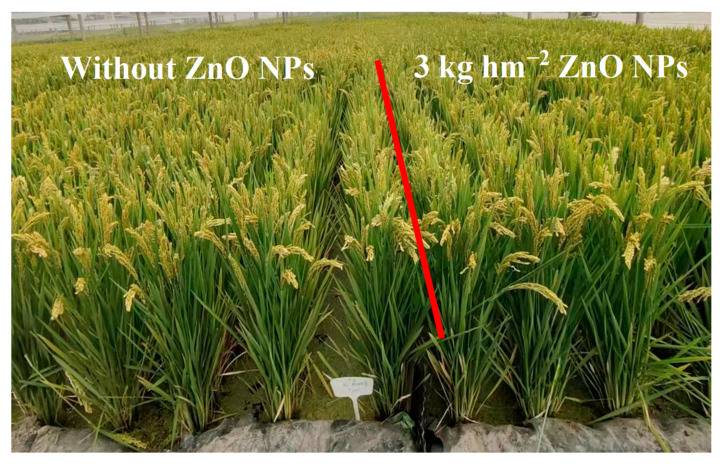
Growth status of field rice during the grain-filling stage after foliar spraying with ZnO NPs at the jointing stage.

**Figure 4 foods-12-03677-f004:**
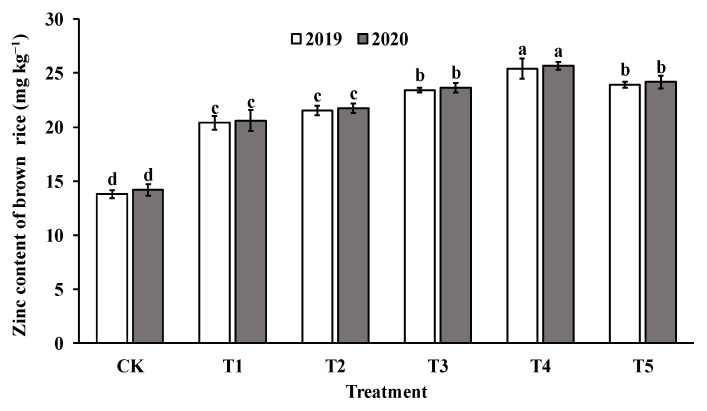
Effect of ZnO NPs concentration on Zn content in brown rice. Values within the same year (2019 or 2020) that are followed by different letters are considered to be significantly different at the 0.05 probability level.

**Figure 5 foods-12-03677-f005:**
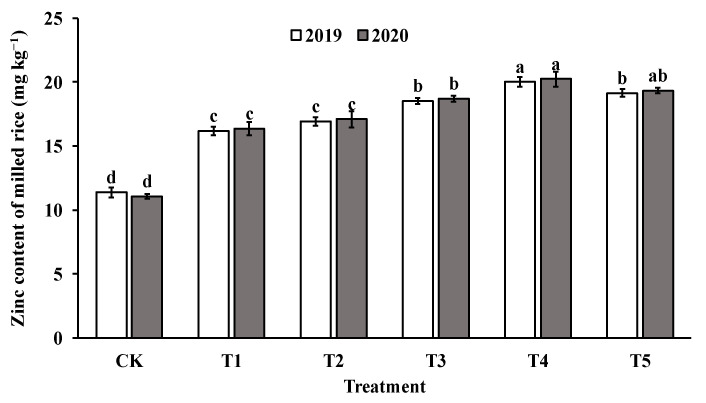
Effect of ZnO NPs concentration on Zn content of milled rice. Values within the same year (2019 or 2020) that are followed by different letters are considered to be significantly different at the 0.05 probability level.

**Figure 6 foods-12-03677-f006:**
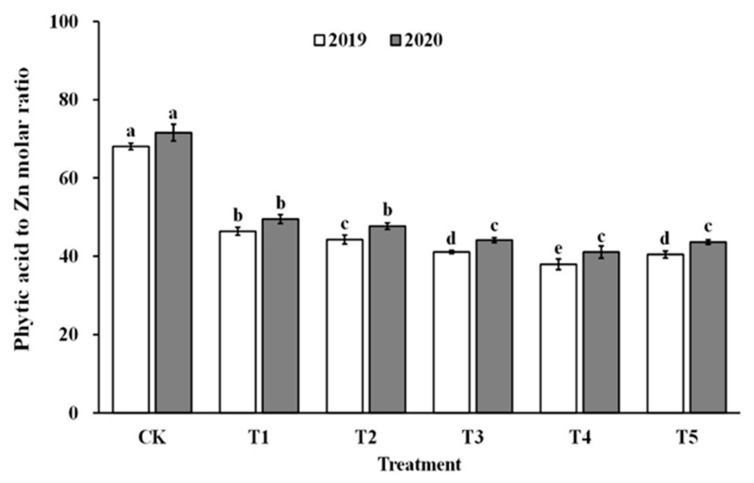
Effect of ZnO NPs concentration on the phytic acid to zinc molar ratio in brown rice. Values within the same year (2019 or 2020) that are followed by different letters are considered to be significantly different at the 0.05 probability level.

**Figure 7 foods-12-03677-f007:**
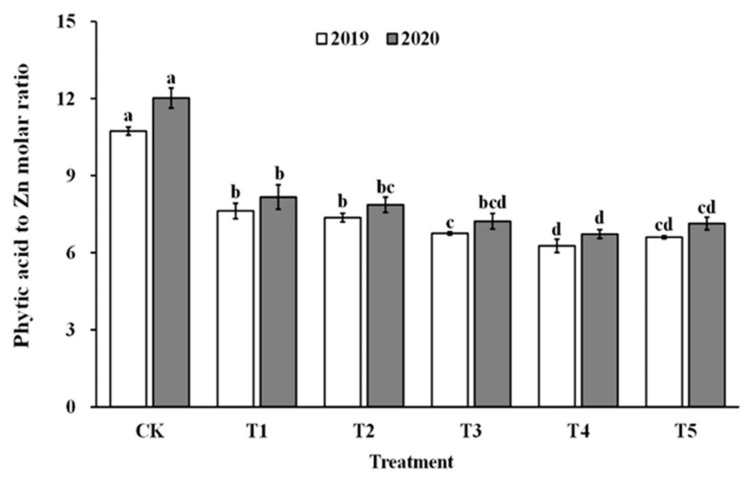
Effect of ZnO NPs concentration on the phytic acid to zinc molar ratio in milled rice. Values within the same year (2019 or 2020) that are followed by different letters are considered to be significantly different at the 0.05 probability level.

**Table 1 foods-12-03677-t001:** Effect of ZnO NP concentration on rice yield and its component.

Year	Treatment	Panicles (×10^6^ hm^−2^)	Spikelets per Panicle	Grain Filling Rate (%)	1000-Grain Weight (g)	Grain Yield (t hm^−2^)
2019	CK	3.54 ± 0.20 a	110.76 ± 8.52 b	90.33 ± 0.60 c	25.33 ± 0.55 b	9.84 ± 0.20 b
	T1	3.56 ± 0.26 a	119.07 ± 3.14 ab	91.30 ± 1.07 c	26.30 ± 0.90 ab	10.09 ± 0.11 ab
	T2	3.55 ± 0.20 a	119.30 ± 4.02 ab	92.35 ± 1.70 bc	26.63 ± 0.19 a	10.14 ± 0.04 a
	T3	3.54 ± 0.20 a	119.37 ± 3.39 ab	92.47 ± 0.57 bc	26.73 ± 0.33 a	10.19 ± 0.06 a
	T4	3.58 ± 0.07 a	119.63 ± 6.18 ab	93.49 ± 0.75 ab	27.05 ± 0.11 a	10.22 ± 0.19 a
	T5	3.55 ± 0.33 a	120.92 ± 2.72 a	94.94 ± 0.43 a	27.08 ± 0.24 a	10.26 ± 0.13 a
2020	CK	3.52 ± 0.07 a	107.82 ± 1.50 b	89.95 ± 0.35 b	25.48 ± 0.20 b	9.97 ± 0.10 b
	T1	3.53 ± 0.14 a	115.63 ± 0.88 ab	92.09 ± 1.42 ab	26.62 ± 0.23 ab	10.18 ± 0.15 a
	T2	3.56 ± 0.18 a	115.69 ± 2.03 ab	92.25 ± 0.25 ab	26.86 ± 0.50 ab	10.23 ± 0.08 a
	T3	3.55 ± 0.25 a	115.74 ± 6.10 ab	92.47 ± 0.86 a	27.20 ± 0.51 a	10.29 ± 0.04 a
	T4	3.52 ± 0.29 a	117.04 ± 4.50 a	92.49 ± 1.39 a	27.24 ± 0.19 a	10.31 ± 0.13 a
	T5	3.61 ± 0.26 a	117.70 ± 1.96 a	93.04 ± 2.01 a	27.32 ± 1.75 a	10.36 ± 0.04 a

Note: Values within the same column for each year (2019 or 2020) that are followed by different letters are significantly different at the 0.05 probability level.

**Table 2 foods-12-03677-t002:** Effect of ZnO NPs concentration on processing quality of rice.

Year	Treatment	Brown Rice Rate (%)	Milled Rice Rate (%)	Head Rice Rate (%)
2019	CK	84.94 ± 0.18 e	75.45 ± 0.18 b	59.26 ± 0.48 b
	T1	85.45 ± 0.12 d	76.12 ± 0.26 a	60.85 ± 0.94 a
	T2	85.68 ± 0.18 cd	76.29 ± 0.09 a	61.44 ± 0.89 a
	T3	85.87 ± 0.14 bc	75.87 ± 0.29 ab	61.69 ± 0.77 a
	T4	86.49 ± 0.17 a	75.49 ± 0.13 b	61.76 ± 0.88 a
	T5	86.06 ± 0.10 b	76.27 ± 0.44 a	61.72 ± 1.15 a
2020	CK	85.23 ± 0.35 b	75.55 ± 0.24 b	59.27 ± 0.88 b
	T1	85.48 ± 0.13 ab	75.76 ± 0.33 ab	60.57 ± 0.73 ab
	T2	85.56 ± 0.07 ab	75.92 ± 0.36 ab	60.83 ± 0.88 ab
	T3	85.64 ± 0.18 ab	76.12 ± 0.26 ab	61.15 ± 0.75 ab
	T4	86.19 ± 0.58 a	76.32 ± 0.14 a	61.28 ± 0.70 a
	T5	86.18 ± 0.39 a	76.40 ± 0.40 a	61.38 ± 0.77 a

Note: Values within the same column for each year (2019 or 2020) that are followed by different letters are considered to be significantly different at the 0.05 probability level.

**Table 3 foods-12-03677-t003:** Effect of ZnO NPs concentration on appearance quality of rice.

Year	Treatment	Length/Width	Chalkiness Grain Rate (%)	Chalkiness Degree (%)
2019	CK	1.669 ± 0.020 b	57.56 ± 1.04 a	18.14 ± 1.880 a
	T1	1.694 ± 0.015 ab	52.28 ± 2.76 b	17.74 ± 1.12 ab
	T2	1.709 ± 0.007 a	51.07 ± 3.24 b	17.07 ± 2.10 ab
	T3	1.713 ± 0.012 a	51.83 ± 0.51 b	16.82 ± 0.47 ab
	T4	1.703 ± 0.004 a	43.95 ± 1.54 c	14.73 ± 0.68 b
	T5	1.702 ± 0.005 a	45.98 ± 1.59 c	14.82 ± 0.91 b
2020	CK	1.679 ± 0.004 b	56.81 ± 3.40 a	20.42 ± 1.84 a
	T1	1.695 ± 0.013 ab	53.27 ± 2.73 ab	18.73 ± 1.14 ab
	T2	1.705 ± 0.004 a	49.47 ± 4.52 abc	16.01 ± 1.68 cd
	T3	1.709 ± 0.007 a	47.88 ± 4.01 bc	17.97 ± 0.67 abc
	T4	1.710 ± 0.008 a	43.88 ± 1.10 c	14.73 ± 0.88 d
	T5	1.707 ± 0.010 a	49.53 ± 2.66 abc	16.97 ± 1.95 bcd

Note: Values within the same column for each year (2019 or 2020) that are followed by different letters are considered to be significantly different at the 0.05 probability level.

**Table 4 foods-12-03677-t004:** Effect of ZnO NPs concentration on nutrition quality of rice.

Year	Treatment	Protein Content (%)	Amylose Content (%)
2019	CK	7.27 ± 0.50 a	11.36 ± 0.68 a
	T1	7.30 ± 0.30 a	11.50 ± 0.72 a
	T2	7.40 ± 0.14 a	11.63 ± 0.59 a
	T3	7.23 ± 0.05 a	11.22 ± 0.24 a
	T4	7.20 ± 0.01 a	11.14 ± 0.44 a
	T5	7.20 ± 0.15 a	11.04 ± 0.28 a
2020	CK	7.27 ± 0.12 a	12.43 ± 0.30 a
	T1	7.30 ± 0.14 a	11.98 ± 0.60 ab
	T2	7.17 ± 0.09 a	12.19 ± 0.74 ab
	T3	7.10 ± 0.16 a	12.38 ± 0.45 a
	T4	7.07 ± 0.12 a	11.66 ± 0.29 ab
	T5	7.07 ± 0.21 a	11.21 ± 0.30 b

Note: Values within the same column for each year (2019 or 2020) that are followed by different letters are considered to be significantly different at the 0.05 probability level.

**Table 5 foods-12-03677-t005:** Effect of ZnO NPs concentration on taste quality of rice.

Year	Treatment	Taste Value	Appearance Value	Hardness Value	Viscosity Value	Balance Value
2019	CK	71.27 ± 1.68 b	6.67 ± 0.26 b	6.70 ± 0.14 ab	7.20 ± 0.22 b	6.70 ± 0.28 b
	T1	78.17 ± 0.63 ab	7.77 ± 0.12 a	6.20 ± 0.08 b	8.43 ± 0.17 a	7.87 ± 0.12 a
	T2	79.17 ± 1.53 a	7.97 ± 0.24 a	6.37 ± 0.05 ab	8.43 ± 0.21 a	8.03 ± 0.26 a
	T3	78.97 ± 1.91 a	7.87 ± 0.29 a	6.07 ± 0.24 b	8.57 ± 0.21 a	7.97 ± 0.29 a
	T4	82.37 ± 2.22 a	8.07 ± 1.01 a	6.03 ± 0.79 b	8.60 ± 0.86 a	8.13 ± 0.97 a
	T5	79.03 ± 2.95 a	7.97 ± 0.41 a	7.00 ± 0.17 a	8.37 ± 0.33 a	8.03 ± 0.42 a
2020	CK	73.97 ± 1.41 b	7.40 ± 0.16 b	6.27 ± 0.51 ab	7.50 ± 0.24 c	7.33 ± 0.12 b
	T1	79.00 ± 2.16 ab	7.93 ± 0.31 ab	6.03 ± 0.19 b	8.40 ± 0.22 b	8.03 ± 0.31 a
	T2	79.33 ± 2.62 a	8.00 ± 0.36 ab	6.07 ± 0.12 b	8.47 ± 0.31 ab	8.10 ± 0.36 a
	T3	80.67 ± 2.05 a	8.17 ± 0.34 a	6.00 ± 0.14 b	8.63 ± 0.12 ab	8.30 ± 0.29 a
	T4	81.67 ± 1.70 a	8.33 ± 0.21 a	5.90 ± 0.08 b	8.70 ± 0.14 ab	8.43 ± 0.16 a
	T5	81.33 ± 2.62 a	8.23 ± 0.26 a	6.80 ± 0.51 a	9.17 ± 0.58 a	8.30 ± 0.29 a

Note: Values within the same column for each year (2019 or 2020) that are followed by different letters are considered to be significantly different at the 0.05 probability level.

**Table 6 foods-12-03677-t006:** Effect of ZnO NPs concentration on RVA parameters.

Year	Treatment	Peak Viscosity (cP)	Trough Viscosity (cP)	Final Viscosity (cP)	Breakdown (cP)	Setback (cP)
2019	CK	2655.00 ± 32.14 b	2004.67 ± 97.68 a	2478.00 ± 91.91 a	650.33 ± 61.34 b	−177.00 ± 20.06 a
	T1	2725.67 ± 14.57 a	1846.00 ± 55.47 ab	2388.67 ± 38.68 a	879.67 ± 43.53 a	−337.00 ± 25.82 ab
	T2	2742.00 ± 27.50 a	1758.67 ± 74.81 ab	2313.33 ± 60.48 a	983.33 ± 74.77 a	−428.67 ± 63.57 b
	T3	2747.00 ± 18.52 a	1753.33 ± 11.59 b	2386.33 ± 14.74 a	993.67 ± 27.65 a	−360.67 ± 11.24 ab
	T4	2696.00 ± 41.94 ab	1712.67 ± 22.55 b	2351.33 ± 41.04 a	983.33 ± 20.43 a	−344.67 ± 30.01 ab
	T5	2727.33 ± 17.79 a	1830.33 ± 69.74 ab	2358.33 ± 80.16 a	897.00 ± 47.22 a	−369.00 ± 62.45 b
2020	CK	2708.00 ± 42.58 ab	1937.33 ± 64.85 a	2430.67 ± 154.05 a	770.67 ± 36.26 b	−277.33 ± 28.00 a
	T1	2683.33 ± 15.95 ab	1706.33 ± 28.57 b	2239.33 ± 35.57 b	977.00 ± 37.27 a	−444.00 ± 44.03 b
	T2	2602.67 ± 10.02 c	1604.33 ± 51.16 bc	2168.00 ± 50.69 bc	998.33 ± 60.93 a	−434.67 ± 60.01 b
	T3	2661.33 ± 14.47 b	1652.00 ± 42.00 bc	2235.67 ± 32.25 bc	1009.33 ± 40.02 a	−425.67 ± 20.40 b
	T4	2732.67 ± 36.35 a	1714.33 ± 43.36 b	2259.33 ± 11.37 b	1018.33 ± 52.37 a	−473.33 ± 32.03 b
	T5	2559.67 ± 12.52 c	1544.00 ± 29.51 c	2118.67 ± 34.79 c	1015.67 ± 31.47 a	−441.00 ± 37.04 b

Note: Values within the same column for each year (2019 or 2020) that are followed by different letters are considered to be significantly different at the 0.05 probability level.

## Data Availability

The raw data supporting the conclusions of this article will be made available by the authors, without undue reservation.

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
