# Peer review of "Foliar Spraying of ZnO Nanoparticles Enhanced the Yield, Quality, and Zinc Enrichment of Rice Grains"

_foods, 2023, doi:10.3390/foods12193677_

Round 1

Reviewer 1 Report

Comments and Suggestions for Authors

I have gone through the work "Foliar spraying of ZnO nanoparticles enhanced the yield, qual- 2 ity, and zinc enrichment of rice grains simultaneously" by Wang et al.. They claim that ZnO nanoparticle spraying during the gestation stage 318 of rice plants could have multiple benefits, including increasing yield, improving quality, and achieving effective zinc enrichment in rice grain edible parts. The flow of writting pattern is not uniform and not organised. 

Author Response

ResponseThank you for your feedback and for reviewing the work "Foliar spraying of ZnO nanoparticles enhanced the yield, quality, and zinc enrichment of rice grains simultaneously" by Wang et al. We appreciate your input regarding the flow and organization of the writing pattern. We will carefully consider your comments and work on improving the uniformity and organization of the manuscript to enhance its clarity and readability. We have thoroughly revised the manuscript to meet your requirements. Please review the revised manuscript attached in the appendix. Your insights are valuable in helping us refine the presentation of our research findings.

Reviewer 2 Report

Comments and Suggestions for Authors

Title of Manuscript: Foliar spraying of ZnO nanoparticles enhanced the yield, quality, and zinc enrichment of rice grains simultaneously:
The present research elucidates the effect of ZnO nanoparticles on grain yield and yield quality parameters in rice, including Zn enrichment in rice grains. The manuscript fits in as a potential study .I suggest taking editorial assistance from language experts.
Abstract:
* The abstract should be more crisp and concise. Rectify the grammatical errors throughout.
* Give problem statement in a better way and add objectives in precise manner
* Please briefly add the methodology
* Line 16= gestation stage “add day after sowing”.
* Line 17= filled grain rate replace with grain filling rate.
* Line 18= 1000-grain weight, leading to a significant improvement in rice yield as compared to what ?
* Future perspective of current study should be added in abstract.
* I suggest you rewrite the whole abstract and differentiate the results in %.
* Keywords should be in alphabetical order.
Introduction:
* Introduction should be revised and re-written.
* Add 3 more lines of the significance of rice in human diet.
* Drastic typographical errors are present in the Introduction and should be rectified.
* Check reference style of the journal and abide to it strictly.
* Authors should specifically mention novelty of the research.
Revisit the novelty of your study. Write hypothesis before the aim of study.
Material and methods:
* Several points need to be clarified. Even treatment description is not Many statements are
confusing. Statistically analysis is also missing.
* Better to add meteorological data in the form of a graph.
* Rewrite the whole section. Make things precise and clear.
* Spray water was used as a control (CK), you mean distilled water spray?
* Mention Number of days after sowing and also mention the exact time of sampling and harvesting.
Results and discussion:
* Write interaction effect 1st and then discuss further.
* In results, better to add the numeric description of results of each trait
* Language needs substantial improvement. There are many grammatical and typo mistakes in the results and discussion.
* Tables must be self-explanatory. Add all the abbreviation in the table.
* Please keep in mind the difference between a review of literature and discussion. In discussion, you should be discussing your own results rather than citing another scientist.
* I also suggest you to add correlation matrix among results.
Conclusion
* Just report the key findings. It should not be a detailed summary of the work done.

Comments on the Quality of English Language

English is better in this study but moderate improvement will be valuable.

Author Response

Thank you for your comments.

Reviewer 3 Report

Comments and Suggestions for Authors

Comments to the authors

 In this review, the authors reported the effects of foliar spray of ZnO NPs fertilizers on rice yield and its components, processing, appearance and nutrition quality of rice by field experiments for 2 years. I think this experiment is essential to demonstrate the impact of this fertilizer on rice yield and quality. The authors’ way of experimental design and presenting data are simple and easy to understand for the readers.

To be a more understandable and qualified paper, the following comments should be addressed and the manuscript should be thoroughly revised.

- As much as I know, the terms such as “gestation stage” are not often used.  Please add explanation to this term or use more common term both in Abstract and throughout MS.

-      - If the rice experiments in the fields can be shown, it will be more realizing and attractive MS to the readers.

-Line 19: “NPs increased the rate of the brown rice, milled rice…..”

I do not understand well. Do the authors mean NPs increased the numbers of brown rice, milled rice, ….? Please revise it to be more understandable.

-      Line 20: Explain more about “rice breakdown value” inside MS.

-      Compared to other Zn fertilizers including ZnSO4, Zn-EDTA, Zn-Gly, this ZnO NPs fertilizer is cheaper?

-      Line 328: it is concluded that 3-6 kg hm2 is the optimal amount of ZnO NPs to be sprayed. What is the estimated cost of this fertilizers for farmers?

 -      And can it be produced a large amount for practical application?

-Line 237 and Table 7: add the meaning of “Setback value” in MS?

It would be better to explain the meanings of Specific terms in the manuscripts for the readers from various background.

-      In this experiment, the authors used Japonica rice. How about NPs effects on other type such as indica, glaberrima?

-      All results are shown as Tables. I cannot find any Figures in the MS. Especially, Zn contents in brown and polished rice are important information. The authors should resubmit the missing results.

-      Did the authors measure other important minerals of brown and polished grains?

Line 16, 109, 111, etc., improve -2 to superscript.

-       

Comments on the Quality of English Language

Sufficient

Author Response

Thank you for your comments.

Reviewer 4 Report

Comments and Suggestions for Authors

Line 16= i am conserned about these concentrations. Can author explain how nanoparticles were added in how much water or etc?

Line 17-18=please write about increased percentage of your result. 

Which concentration is best? 

Line 108=please clarify your experimental design. 

Line 115-ZnoNps concentrations mentioned here are different in formulations as in abstract. Please confirm. 

Did the authors purchased nanoparticles or Synthesized? If Synthesized please provide methodology and all characterization techniques. Please also provide characterization photo in case of purchased? 

Also provide the mode of syntheses either chemical synthesis or green syntheses 

Comments on the Quality of English Language

Extensive English editing is needed 

Author Response

Thank your for your comments.

Reviewer 5 Report

Comments and Suggestions for Authors

1- Some numbers in the manuscript should be superscript like K-1

2- In all tables, a line should separate the years and the note under tables should be correct and mention the separate year:

Note: Values within the same column for each year, followed by different letters are significantly different at the 0.05 probability level.

3- Why did not you analyze the data as a combined analysis of variance?

Author Response

Thank you for your comments.

Round 2

Reviewer 4 Report

Comments and Suggestions for Authors

Thank you for your response. I am not satisfy with comment 1

Author claimed that  "To prepare the ZnO NP s solutions, 4.5, 9.0, 18.0, 36.0, and 72.0 g in I L .

when I asked already about best concentration then author replayed "3 to 6 kg hm-2." can author explain how it possible that a concentration is best which not used. 

The best concentration should be among these concentration 4.5, 9.0, 18.0, 36.0, and 72.0 g. 

My second objection is all applied concentrations are very high. Can author provide me about any previous study that used these concentrations or similar?

Comments on the Quality of English Language

English editing is needed.

Author Response

#Reviewer4

Thank you for your response. I am not satisfying with comment 1.

Comment 1: Author claimed that "To prepare the ZnO NPs solutions, 4.5, 9.0, 18.0, 36.0, and 72.0 g in 1 L. When I asked already about best concentration then author replayed "3 to 6 kg hm-2." Can author explain how it possible that a concentration is best which not used. The best concentration should be among these concentration 4.5, 9.0, 18.0, 36.0, and 72.0 g. 

Response: Thank you for your comment. The optimal amount of ZnO NPs spray during the rice jointing period was estimated to fall within the range of 18.0 to 36.0 g L-1.

Comment 2: My second objection is all applied concentrations are very high. Can author provide me about any previous study that used these concentrations or similar?

ResponseThank you for your comment. Based on our preliminary spraying investigation, we initially applied various dosages of ZnO nanoparticles (0.5, 1, 2, 4, and 6 kg hm-2) to the rice plants. The outcomes consistently indicated an upward trend in grain zinc content with increasing ZnO NP dosages (refer to Table S1). Encouraged by these initial results, we decided to expand the range of ZnO nanoparticle dosages for a more comprehensive assessment. This expansion aimed to pinpoint the optimal dosage for our specific study. Consequently, through this systematic process, we identified the most effective dosage that significantly enhanced the zinc content in the rice grains. This dosage was subsequently employed in our research.

Table S1: Effects of foliar spraying various dosages of ZnO NPs at jointing stage on zinc content of rice grain.

Treatment

Zinc content (mg kg-1)

CK

13.20±0.23f

T1

15.10±0.11e

T2

17.51±0.08d

T3

19.22±0.19c

T4

22.46±0.10b

T5

24.43±0.07a

Note: Values within the same column followed by different letters are significantly different at the 0.05 probability level.
